# Multifunctional sequence-defined macromolecules for chemical data storage

Steven Martens[1], Annelies Landuyt[1], Pieter Espeel[1], Bart Devreese [2], Peter Dawyndt [3] & Filip Du Prez [1]

Sequence-defined macromolecules consist of a defined chain length (single mass), end-groups, composition and topology and prove promising in application fields such as anti-counterfeiting, biological mimicking and data storage. Here we show the potential use of multifunctional sequence-defined macromolecules as a storage medium. As a proof-of-principle, we describe how short text fragments (human-readable data) and QR codes (machine-readable data) are encoded as a collection of oligomers and how the original data can be reconstructed. The amide-urethane containing oligomers are generated using an automated protecting-group free, two-step iterative protocol based on thiolactone chemistry. Tandem mass spectrometry techniques have been explored to provide detailed analysis of the oligomer sequences. We have developed the generic software tools Chemcoder for encoding/decoding binary data as a collection of multifunctional macromolecules and Chemreader for reconstructing oligomer sequences from mass spectra to automate the process of chemical writing and reading.

---

[1] Department of Organic and Macromolecular Chemistry, Polymer Chemistry Research Group, Centre of Macromolecular Chemistry (CMaC), Ghent University, Krijgslaan 281 S4bis, 9000 Ghent, Belgium. [2] Department of Biochemistry and Microbiology, Laboratory for Protein Biochemistry and Biomolecular Engineering, Ghent University, K.L. Ledeganckstraat 35, 9000 Ghent, Belgium. [3] Department of Applied Mathematics, Computer Science and Statistics, Ghent University, Krijgslaan 281 S9, 9000 Ghent, Belgium. Correspondence and requests for materials should be addressed to F.D.P. (email: Filip.DuPrez@UGent.be)

Reliable data storage, already in the zettabyte ($10^{21}$) range and increasing, is one of the largest technological challenges of the digital age[1]. While current conventional storage devices are still able to cope with this increasing demand, they occupy large floor areas in data centres, depend on ever rarer elements and require a great deal of energy, a resource already being stretched to the edge of current capacity. Encoding data at the (macro)molecular level could overcome these drawbacks, because physical maintenance is negligible, storage densities can be dramatically increased and sources of elements (C, H, N, O) as constituents of the information-containing macromolecules are highly abundant.

DNA, carrier of genetic information and arguably nature's largest biopolymer, has already been used as a macromolecular carrier of information, able to archive[2–4], manage (DNA hard disk)[5] and encrypt data[6–8] easily retrieved by well-established read-out tools[9]. Moreover, immense storage densities can be achieved, i.e. $10^6$ times more information per mm³ than in hard disk or flash memories[10]. For example, 200 megabytes of data, including a high-definition music video and 100 books, were recently stored on DNA that contained more than 1.5 billion base pairs[11].

Although the encoded information can be copied by DNA replication, susceptible to errors, DNA holds serious practical issues related to long-term stability and synthetic scalability[12]. Indeed, DNA is sensitive to both hydrolysis and other degradation reactions, such as deamination and dimerization. These issues could be overcome with synthetic sequence-defined polymers if the backbone and side chains are chosen wisely. The structure of DNA is also quite complex, and the four-letter nucleotide code that makes up its 'alphabet' is limited compared to the vast diversity of synthetic building blocks. Another important issue hampering the large-scale use of DNA is the limited availability, the latter mostly connected with the scarceness of biologically available phosphorus in nature[13]. Although recent research indicates that DNA is more stable than flash memory and that the amount of silicon might not be able to cope with the production of chips, it should be emphasized that, compared to phosphorus, silicon is 300 times more easily available on earth and can be retrieved from more accessible minerals[3]. While producing DNA on a large enough scale is not feasible, it has recently been proven that sequence-defined polymers can be made on multigram scale with cheap compounds, which shows the potential for further industrial upscaling[14–16]. In addition, many types of polymers are known to remain fairly stable over a very long period of time (decennia to centuries), and their cost is significantly lower than that of DNA production, even for tailor-made structures. Polymer chemists have recently realized that the potential of data storage is not restricted to DNA: recent progress in the field indicates that sequence-defined macromolecular structures are equally applicable for data storage purposes[17,18]. Therefore, they have been developing different methodologies to achieve full control over the primary structure, i.e. the order of monomers in a sequence of synthetic macromolecules[14,16,19–36]. Moreover, synthetic carriers of digital information can have significantly simplified structures regarding backbone and chirality. In fact, the simplest constructs are atactic, sequence-defined binary strings[1]. Compared to DNA, these polymers, devised as digital information carriers, have very simple molecular structures. However, in theory they are more robust, and therefore potentially constitute the basis for future data storage technologies.

Various synthetic approaches have been developed to make sequences that can be easily read with well-known technologies such as tandem mass spectrometry[20,37–40]. Lutz and co-workers, for instance, stored digital information on sequence-defined oligourethanes[26,27], oligo(triazole amide)s[28,29], poly(phosphodiester)s[30,31], oligo(alkoxyamine amide)s[32,33] and oligo(alkoxyamine phosphodiester)s[34,35]. In these sequences, digital information is encoded using two monomers, resulting in the binary representation commonly used by modern computer systems. While these approaches usually store data in binary form, recent synthetic approaches have attempted to expand the number of functionalities that can be placed in the backbone or side-chain[16,20,40–42]. After all, the amount of data that can be stored in macromolecular chains depends on both the chain length and the number of different building blocks, which determine the base of the positional numeral system[43]. For example, with 20 different building blocks (base-20), up to $8 \times 10^3$ trimers and $2.56 \times 10^{10}$ different octamers can be made, and while only 32 different pentamers can be made with 2 different building blocks (base-2 = binary), $3.2 \times 10^6$ pentamers are theoretically possible through the use of 20 building blocks. Thus, the use of sequence-defined structures with a large diversity of functionalities would allow for compact data storage on short macromolecular chains. The first examples of using multiple functionalities for data storage have recently been reported by the groups of Lutz[35] and Meier[44].

We recently reported on two approaches for the straightforward synthesis of multi-functional sequence-defined oligo(amide-urethane)s by making use of thiolactone (Tla) chemistry[45–47]. Although a range of chemical functionalities could easily be inserted with both approaches, the one using acrylates, instead of amines, to introduce side-chain functionalities was more advantageous[46]. It resulted in longer high-purity sequences and made use of an automated protocol (Fig. 1). Two different linkers, a Tla-containing alcohol and an acid, were used to connect the thiolactone moiety and solid support[45,46].

For peptide chemistry and synthetic sequences, examples can already be found of computational algorithms that revolutionized sequence-order reading, database building and de novo identification[48–50].

Our research was inspired by the controlled fragmentation of the prepared sequence-defined oligomers and the time-consuming interpretation of the MS/MS-spectra to develop an automated sequencing tool, called Chemreader. The algorithm is first tested and improved by decoding a sentence written on oligo(amide-urethane)s. Then, the power of this algorithm and protocol is further exemplified by encoding and decoding a QR (Quick Response) code starting from short, multifunctional macromolecular structures. Another algorithm, called Chemcoder, is developed to automate the translation of binary data, such as the QR code, to oligomers and vice versa (Fig. 1). To write both human- and machine-readable information on oligomeric structures, more than 15 different side-chain functionalities are used.

## Results
**Read-out of the oligomers**. In order to determine on the one hand the fragmentation behaviour of the oligomers and on the other hand the most suitable mass technique for the oligo(amide-urethane)s analysis (electrospray ionization (ESI) or matrix-assisted laser desorption/ionization (MALDI) tandem mass spectrometry), a pentamer Z5 was first prepared starting from an acid linker and analysed (Supplementary Tables 1, 2 and Supplementary Figures 14–17). The fragments generated in the collision cell of these mass spectrometers mainly resulted from a controlled fragmentation on the urethane bond. As can be seen in Fig. 2, the sequence can be fully read from left to right and vice versa. In terms of potential mass range analysis for longer

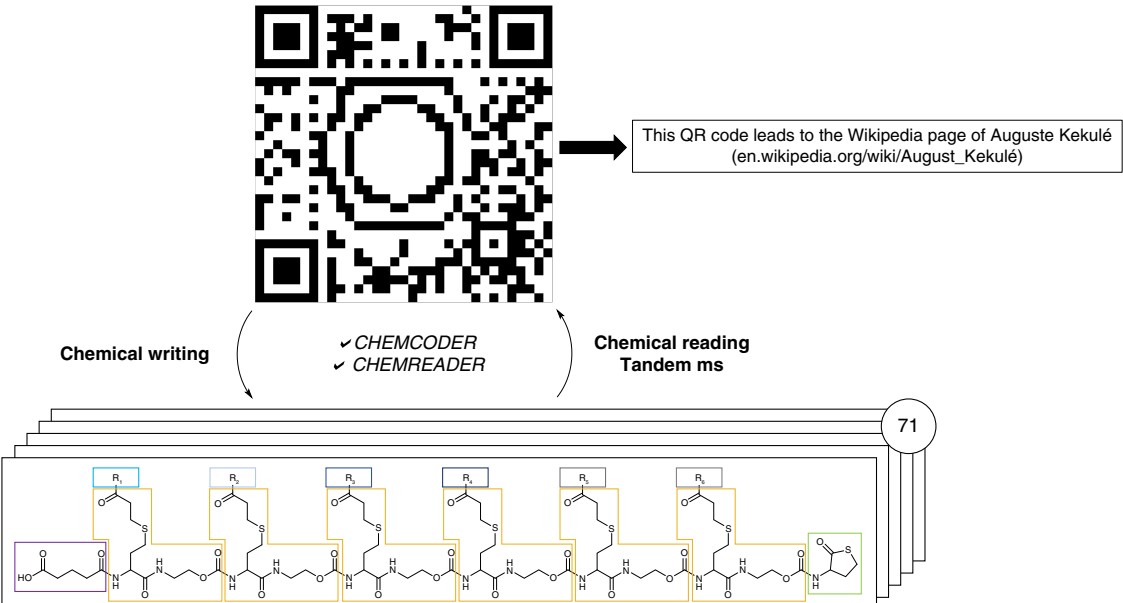

**Fig. 1** Schematic representation of the QR code conversion. A QR code including a benzene structure, encoding the URL of the Wikipedia page of August Kekulé, who was the first to understand the structure of benzene and made a proposal for its structure, is translated to and written on 71 different sequences. Translation is done using the Chemcoder algorithm. The sequences are read afterwards by means of tandem MS and the Chemreader algorithm. Given these sequences, Chemcoder is able to reconstruct the original QR code. Different building blocks in the sequence are highlighted: start fragment (purple box); backbone (yellow boxes), stop fragment (green box) and the functionalities (grey and blue boxes)

sequences, we decided to continue with MALDI-TOF/TOF MS/MS. Although both positive and negative ion mode proved to work in the past for a variety of sequence defined oligomers[51], only positive mode was used here because the signal-to-noise ratio of MALDI-TOF MS signals is typically better in this mode. While it has been shown that the same synthetic platform also allows for modification of the oligomer backbone, the level of functionalization has been restricted to the side-chain in order to reduce the complexity of the resulting mass spectra (vide infra)[46].

Following this initial study, six hexamers that were previously prepared on an automated synthesizer starting from an alcohol linker[46] (H1-H6) were also deciphered with MALDI-MS/MS (Supplementary Table 3–8 and Supplementary Figures 1, 18–35). These hexamers were built with benzyl-, butyl- or tetrahydrofurfuryl acrylate and contain repetitions in their sequences. Although H2 and H5 have the same mass, the order of their sequence could be easily determined and thus they could be differentiated unambiguously. For sequence H5, a more detailed analysis of the MS/MS spectrum was performed (Supplementary Figure 32).

**Development of Chemreader algorithm**. Once we had proven the easy read-out of these sequences, we explored their potential to store data and developed an algorithm (Chemreader) that automates the read-out process. Pentamer Z5 was used for the initial development of Chemreader. The algorithm uses both the masses of the collection of functionalities and the length of the monomer sequence as input parameters. In a first step, the program generates all possible fragments that could possibly be formed. Subsequently, it searches for matching masses that are obtained after MS/MS analysis. Finally, fragments are combined to reconstruct the original sequence. If we inspect pentamer Z5 in more detail (Fig. 2), fragmentation on the urethane bonds leads to the fragments necessary to perform the automated sequence analysis with the Chemreader algorithm. In all cases, both the

start-containing fragment (left fragment with the acid linker) and the stop-containing fragment (right fragment with the thiolactone ring) are present in the spectrum. Presence of these two fragments makes it easy for the algorithm to unambiguously translate the MS/MS spectrum into the exact pentamer structure. The Chemreader algorithm has linear time complexity in the length of the polymers and the number of building blocks (octamers with a 20-character alphabet are resolved in the order of milliseconds on a standard laptop). A more detailed explanation of the algorithm can be found in the Supplementary Methods (Supplementary Figures 2, 3).

**Writing and reading human-readable data**. Next, we attempted to write the question TO WRITE OR NOT TO WRITE ON OLIGOS? on short oligomers. For this, the eight different words are converted into individual oligomers, using acrylates as a chemical alphabet to represent the individual characters. Comparable to previous research in which mass tags were added to oligomers to indicate the position of a letter in a word[52], the position of the words in the sentence has been encoded to enable the reconstruction of the words in the correct order. As a result, the sentence is actually encoded as 1TO 2WRITE 3OR 4NOT 5TO 6WRITE 7ON 8OLIGOS? The sentence was written twice using the two different linkers, showing the versatility of the α-end groups used for writing the oligomers (Supplementary Table 10–25 and Supplementary Figures 36–83). The acrylates (19 in total, each with a different mass) correspond to the different letters, numbers and the question mark in the sentence (Supplementary Table 9, Supplementary Figures 4–13 and Fig. 3).

Decoding the sentence requires knowledge about the alphabet (acrylates used), the number of words and the length of each word. Each word can be analysed separately. Given this information, Chemreader can reconstruct the original sentence. Only for the word 8OLIGOS?, one peak corresponding to the smallest fragment was absent. However, due to the redundancy in

Chemical formula: $C_{86}H_{133}N_{11}NaO_{30}S_6$
Exact mass: 2014.74418
Molecular weight: 2016.40677

Exact mass: 1878.62
Molecular weight: 1880.17

Exact mass: 459.16
Molecular weight: 459.51

Exact *m/z*: 749.25
Molecular *m/z*: 749.85

Exact *m/z*: 1109.39
Molecular *m/z*: 1110.27

Exact *m/z*: 1441.53
Molecular *m/z*: 1442.69

Exact *m/z*: 1853.73
Molecular *m/z*: 1855.23

Exact *m/z*: 1534.59
Molecular *m/z*: 1535.87

Exact *m/z*: 1244.49
Molecular *m/z*: 1245.54

Exact *m/z*: 884.36
Molecular *m/z*: 885.12

Exact *m/z*: 552.22
Molecular *m/z*: 552.70

Methyl unit — Tetrahydrofurfuryl unit — Butyl unit — Isobornyl unit

Butyl unit — Tetrahydrofurfuryl unit — Methyl unit — Benzyl unit

**Fig. 2** Determining the sequence order. Tandem mass analysis (MALDI-MS/MS) of a pentamer Z5 with five different functionalities. In blue the read-out is highlighted from right to left, in purple from left to right. The coloured arrows indicate the mass difference between two mass fragments and the functionality that is responsible for this difference.

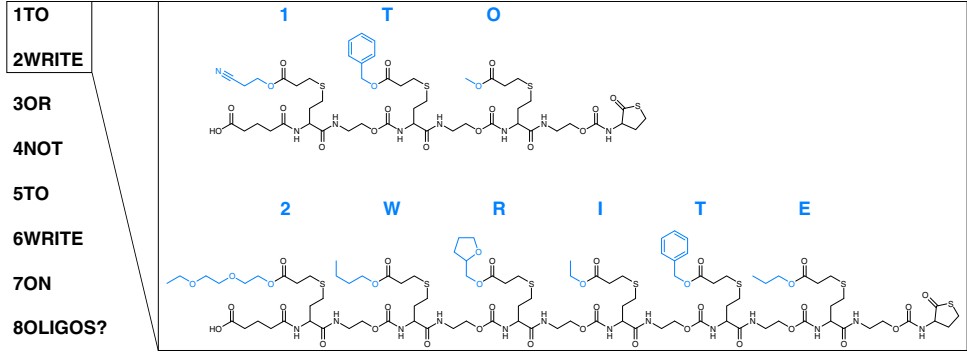

1TO
2WRITE
3OR
4NOT
5TO
6WRITE
7ON
8OLIGOS?

1 T O

2 W R I T E

**Fig. 3** Writing a sentence with sequences. The first two words of the question '1TO 2WRITE 3OR 4NOT 5TO 6WRITE 7ON 8OLIGOS?' in their chemical form. The different functionalities (in blue), introduced via acrylates in the chemical protocol, express a different letter or number

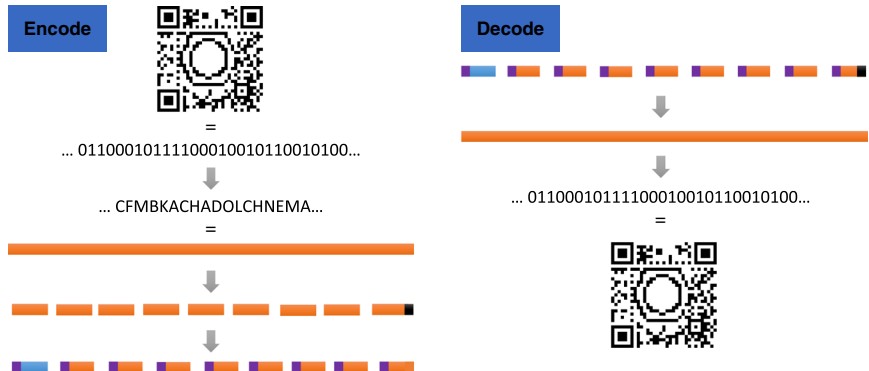

**Fig. 4** Encoding and decoding of the QR code. Encoding scheme (left). The bit string representing the QR code is first translated into a pentadecimal numeral system (base-20). The sequence of 'flags' is then cut into smaller pieces. In a final step, the position of each fragment (purple) and the length of the bit string (blue) is added. The last fragment may be filled with a non-coding spacer (black); Decoding scheme (right). After determination of the sequence of all fragments, they are dereplicated, sorted, trimmed and glued together. Finally, the sequence of flags is converted into the bit string that reconstructs the original QR code

overlapping fragments and the left-right and right-left reconstruction of the data, the octamer could be correctly translated. While encoding a human-readable sentence in sequence-defined polymers provided a first proof-of-principle to demonstrate the power of the Chemreader algorithm, the applied encoding scheme is not scalable to larger text fragments due to variable-length position encoding and to larger alphabet sizes (e.g. ASCII or Unicode) as separate acrylates are needed for all characters in the alphabet.

**Writing and reading of machine-readable data.** A second and more ambitious challenge was the synthesis and analysis of different oligomers to encode a $33 \times 33$ QR code, corresponding to a square grid containing 1089 pixels. With the ever-increasing use of smartphones, QR codes have become a simple way of communicating short messages. In producing a sample QR code that encodes the URL of the Wikipedia page of August Kekulé[53], we took advantage of the redundancy built into these codes—for error correction purposes—to embed a visual representation of the benzene ring. Kekulé was the first to understand the structure of benzene and made a proposal for its structure (1865) during his stay at Ghent University (1858–1867).

The black and white dots in a QR code represent bits (0 and 1) in the binary numeral system. As such, a QR code is nothing more than a two-dimensional bit string. To achieve the goal of encoding the QR code in sequence-defined polymers, the bit string was converted into a sequence of functionalities. To automate the process of encoding and decoding bit strings as collections of oligomers, a software tool called Chemcoder was developed.

The general outline of the Chemcoder algorithm is schematically represented in Fig. 4. The encoding of a QR code bit string is done in a series of steps. In a first step, the bit string is converted into a sequence of so-called flags (=side-chain functionalities). As this sequence of flags is too long to be encoded in a single oligomer, it is split into short fixed-length fragments. To give the last fragment the same length as the other fragments, it occasionally has to be filled with a non-coding spacer region (black region in Fig. 4). To enable reconstruction of the original bit string from the collection of fragments, an index is added to each fragment (purple region in Fig. 4) as well as the total length of the original bit string (blue region in Fig. 4). Decoding can only be done if the sequence of all the fragments has been determined. In that case, Chemcoder dereplicates the sequenced fragments and sorts them in their

original order based on the index, removes the non-coding index and length regions, and glues the coding sections together into a single bit string, from which the spacer region is trimmed using the encoded length of the original bit string. The resulting bit string corresponds to the original QR code. The Chemcoder algorithm has linear time complexity in the length of the bit string (GB-sized files are converted in the order of milliseconds on a standard laptop).

Apart from the bit string that needs to be encoded into a collection of fragments, Chemcoder needs to be configured with the maximal fragment length and the size of the chemical alphabet (available flags). Depending on these settings, a different number of oligomers must be synthesized: the longer the oligomers and the more flags that can be used, the lower the number of oligomers that needs to be synthesized. We have chosen settings for Chemcoder such that the sample QR code is translated (Fig. 4) into a collection of 71 short oligomers (1 monomer, 11 pentamers and 59 hexamers). The automated protocol developed earlier allows for simultaneous synthesis up to 72 sequence-defined structures, which fits the 71 oligomers that are needed here[46]. To write the QR code, these fragments were synthesized using a library of 15 acrylate monomers (Supplementary Table 26 and Supplementary Figures 84–225), which we have labelled A, B, C… O to make them more human-readable. After obtaining spectra from all 71 oligomers, Chemreader reconstructed all fragments without errors, which were then converted into the original bit string by Chemcoder, yielding the original QR code.

As every oligomer had to be analysed separately, a future challenge would be to combine techniques for the analysis of much more complex samples, in order to guarantee a high data density. An example, well known in the context of peptide analysis, consists of the coupling of liquid chromatography to a tandem ESI-MS/MS equipment to separate different oligomers in the LC dimension and determine the sequence in the tandem MS dimension[54].

The storage capacity of sequence-defined oligomers based on thiolactone chemistry was explored. It is possible for such oligomers to directly contain digital information in a useful way (QR code) while a controlled fragmentation on the urethane bond allowed for an easy read-out of the oligomers. An algorithm, called Chemreader, was developed to facilitate the read-out of these sequences, which allows one to read the information stored within sequence-defined structures in a fast and automated way on a standard laptop. The Chemreader algorithm contributes to

solving the sequence-reading bottleneck of sequence-defined polymers. In order to test the Chemreader algorithm, a sentence in natural language was first successfully written and read, followed by the more ambitious challenge of encoding a 33 × 33 QR code in 71 different, analysed oligomers. Besides, we developed the software tool Chemcoder to quickly encode binary data as a compact collection of oligomer fragments, and *vice versa*. Both algorithms are extremely fast and highly configurable for application on other sets of sequence-defined polymers. A reference implementation is available open source on GitHub (see section Additional Information for URL). We invite other groups to apply them on their own data sets or make any modifications for their own needs.

As the results obtained prove the possibilities for using these mono-disperse, multi-functional oligomers in the field of data storage, this study is another indication for the long-term potential that sequence-defined polymers hold to real-world applications and thus provides further validation for this rapidly developing branch of macromolecular chemistry. Undoubtedly, this will spark further research on the analysis and applicability of sequence-defined polymers worldwide. One of the main research challenges remains the further exploration of non-destructive techniques for the read-out of the sequence order in complex mixtures.

## Methods

**Instrumentation**. $^1$H- and $^{13}$C-NMR (Attached Proton Test, APT) spectra were recorded on a Bruker Avance 300 at 300 MHz and a Bruker Avance 500 at 500 MHz. Chemical shifts are presented in parts per million (δ) relative to DMSO-$d_6$ or CHCl$_3$-$d$ (2.50 ppm or 7.27 ppm in $^1$H- and 39.51 ppm or 77.24 ppm in $^{13}$C-NMR, respectively) as internal standard. All samples were analysed with 2D-NMR techniques (COSY, HSQC and HMBC), which provided a full assignment of the structures. All measurements were performed at 25 °C and ACD/NMR Processor was used for the processing of all data. All spectra including 1D $^1$H and $^{13}$C, 2D COSY, $^1$H-$^{13}$C HSQC and $^1$H-$^{13}$C HMBC were recorded in a standard fashion with pulse programs available in the Bruker library. An Agilent technologies 1100 series LC/MSD system equipped with a diode array detector and single quad MS detector (VL) with an electrospray source (ESI-MS) was used for classic reversed phase LC-MS (liquid chromatography mass spectroscopy) and MS analysis. Analytic reversed phase HPLC was performed with a Phenomenex C18 (2) column (5 μ, 250 × 4.6 mm) using a solvent gradient (0 → 100% acetonitrile in H$_2$O in 15 min) and the eluting compounds were detected via UV-detection (λ = 214 nm). High-resolution mass spectra (HRMS) were collected using an Agilent 6220 Accurate-Mass time-of-flight (TOF) equipped with a multimode ionization (MMI) source. Infrared spectra were recorded with Attenuated Total Reflection (ATR) with a PIKE Miracle ATR unit and a Perkin Elmer FTIR SPECTRUM 1000 spectrometer. IR-software of Perkin Elmer was used for the analysis of the spectra. Automated syntheses were performed on a 72-reactor block INTAVIS MultiPep CF Synthesizer with open 5 mL reaction columns equipped with a vortexing unit (refer to Supplementary Methods, Supplementary Figure 1). The speed of vortexing is 550 rpm. ESI mass spectrometry analysis was performed on a Synapt G1 HDMS mass spectrometer (Waters). Samples were diluted in 50% acetonitrile/0.1% formic acid in water and transferred into a 96-well plate. This plate was loaded into an Advion Triversa Nanomate source. From each sample 3 μL was picked with a conductive peptide tip and moved towards the D-chip plate. Typically, 1.3 V was applied on the chip, spraying the sample in the source area of the mass spectrometer which was used in the Q-TOF mode. Tandem mass spectra were generated by collision induced dissociation using Ar as collision gas at 30 eV collision energy. For MALDI analysis, measurements were performed with trans-2-[3-(4-tert-butylphenyl)-2-methyl-2-propenylidene]malonitrile (DCTB, 30 mg/mL in dichloromethane) as a matrix, Sodium trifluoroacetate (19 mg/mL in acetone) as a cationizing agent, and oligomer samples were dissolved in THF (4 mg/mL). Oligomer solutions were prepared by mixing 10 μL of the oligomer, 1 μL of the salt, and 10 μL of the matrix solution. Subsequently, 0.5 μL of this mixture was spotted on the sample plate, and the spots were dried in air at room temperature. 0.5 μL was spotted on a MALDI plate and loaded into the Sciex 4800 MALDI-TOF/TOF MS instrument equipped with an Nd:YAG laser (200 Hz, 355 nm) controlled by 4000 Series Explorer software version 3.5.3 (Applied Biosystems, Germany). The instrument was operated in positive ion mode with delayed extraction and an acceleration voltage of 20 kV with a grid of 15.6 kV. Fragmentation (MS/MS) was performed in positive ion mode at 1 kV using the no gas option. The 4700 Proteomics Analyser Mass Standard kit (Applied Biosystems, Germany) was prepared according to the manufacturers' recommendation and used for external calibration before analysis (mass to charge

range from 800 to 4000 Da). MS/MS calibration was based on the precursor mass of 1570.677 Da of Glu-fibrinopeptide B. Signals were considered as interpretable if the error in *m/z* was not higher than 0.02 and the signal-to-noise had to be higher than 5.

**Materials**. DMSO-$d_6$ ([2206-27-1], ≥99.8%) and CHCl$_3$-$d$ ([865-49-6], ≥99.8%) were purchased from Euriso-top. Acryloyl chloride ([814-68-6], 96%) was purchased from abcr GmbH. Acetonitrile ([75-05-8], HPLC grade), 1,4-Dioxane ([123-91-1], HPLC grade), and Triethylamine ([121-44-8], 99%) were purchased from Acros Organics. DL-Homocysteinethiolactone hydrochloride ([6038-19-3], 99%) was purchased from Haihang industry (Jinan City, China). Magnesium sulphate hydrate [22189-08-8], ≥99%), Potassium carbonate ([584-08-7], ≥99%) and Sodium bicarbonate ([144-55-8], ≥99.5%) were purchased from Carl Roth. Trifluoroacetic acid ([76-05-1], Peptide grade) and 2-Chlorotrityl chloride resin ([42074-68-0], 100-200 mesh, 1% DVB, 1.6 mmol/g) were purchased from Iris Biotech GmbH. Acetyl chloride ([75-36-5], ≥99%), Bromoacetyl bromide ([598-21-0], ≥98%), Butyl acrylate ([141-32-2], ≥99%), Chloroform ([865-49-6], ≥99.8%), Citronellol ([106-22-9], ≥95%), Dichloromethane ([75-09-2], ≥99.8%), Diethylether ([60-29-7], ≥99.9%), *N*,*N*-Diisopropylethylamine (DIPEA, [7087-68-5], 99%), *N*,*N*-Dimethylformamide ([68-12-2], anhydrous, 99.8%), Ethanolamine ([141-43-5], ≥99%), Ethyl acrylate ([140-88-5], 99%), Glutaric anhydride ([108-55-4], 95%), 1-Heptanol ([111-70-6], 98%), 2-Hydroxyethyl acrylate ([818-61-1], 96%), Isobornyl acrylate ([5888-33-5], technical grade), 2-Mercaptoethanol ([60-24-2], ≥99%), Methanol ([67-56-1], ≥99.9%), Methyl acrylate ([96-33-3], 99%), Phenothiazine ([92-84-2], ≥98%), 1-Propanol ([71-23-8], 99.7%), Propargyl acrylate ([10477-47-1], 98%), Pyridine ([110-86-1], ≥99%), Tetrahydrofuran ([109-99-9], ≥99%) were purchased from Sigma-Aldrich and used without purification, except isobornyl acrylate which was distilled. Benzyl acrylate ([2495-35-4], >97%), 2-Cyanoethyl acrylate ([106-71-8], >95%), Cyclohexyl Acrylate ([3066-71-5], >98%), Dibutyltin dilaurate ([77-58-7], >95%), *N*,*N*-Diethylacrylamide ([2675-94-7], >98%), 2-(Dimethylamino)ethyl Acrylate ([2439-35-2], >98%), 2-Ethoxyethanol ([110-80-5], >99%), 2-(2-Ethoxyethoxy) ethyl Acrylate ([7328-17-8], >98%), 2-Ethylhexyl Acrylate ([103-11-7], >99%), Isoamyl Acrylate ([4245-35-6], >98%), 2-Methoxyethyl Acrylate ([3121-61-7], >98%), 1-Nonanol ([143-08-8], >99%) and Triphosgene ([32315-10-9], >98%) were purchased from TCI and used without purification. Tetrahydrofurfuryl acrylate ([2399-48-6]) was purchased from Polysciences and used without purification. Hydrochloric acid 36% p. (HCl, [7647-01-0]) was purchased from Chem-Lab and used without purification. Solvents (CH$_2$Cl$_2$, CHCl$_3$, DIPEA and pyridine) for the chain extension of sequences, the synthesis of α-isocyanato-γ-thiolactone or the immobilization of functionalized thiolactone linkers were distilled from CaH$_2$ prior to use. Silicagel (ROCC, SI 1721, 60 Å, 40–63 μm) was used to perform preparative column chromatography, eluting with technical solvents. The collected fractions were analysed by thin layer chromatography (TLC-plates, Macherey-Nagel, SIL G-25 UV254). The α-isocyanato-γ-thiolactone, the acid-functionalized and hydroxyl-functionalized thiolactone linker, and 3,7-dimethyloct-6-en-1-yl acrylate (citronellyl acrylate) were synthesized according to literature procedures[45,46,55,56].

**Experimental procedures**. Detailed experimental procedures are described in the Supplementary Methods and are accompanied with reaction schemes when appropriate.

**Code availability**. Both the algorithms can be found at https://github.com/chemstore. The individual algorithms can be found at https://github.com/chemstore/chemcoder and https://github.com/chemstore/chemreader.

## Data availability

All relevant data are available within the paper and its Supplementary Information. The algorithms can be found at https://github.com/chemstore. All other data are available from the authors upon request.

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

## Acknowledgements

Jan Goeman and Isabel Vandenberghe are acknowledged for the MS measurements. S.M. thanks the Research Foundation-Flanders (FWO). This work was supported by FWO through EOS-project 30650939. B.D. acknowledges the Hercules foundation (AUGENT09).

## Author contributions

All authors contributed to discussion and evaluation of the results at all stages. F.D.P, P.E. and S.M. conceived and designed the experiments. S.M. performed the experiments and prepared all figures. A.L. and P.D. developed the algorithms. S.M. and A.L. wrote the manuscript in close discussion with B.D., F.D.P., P.D. and P.E.

## Additional information

**Competing interests:** The authors declare no competing interests.

