## [Peer Review File · Nature Communications]

Reviewers' comments:

Reviewer #1 (Remarks to the Author):

Du Prez et al. report an interesting and novel system to store data in molecules. The manuscript is at the forefront of molecular data storage and sequence defined macromolecules, but also addresses question from other scientific disciplines and is thus well suited for the readership of Nature Communications. Storage capacity is one of the highest reported yet (maybe compare to a very recent manuscript: <https://doi.org/10.1016/j.eurpolymj.2018.04.038>) per repeat unit and is further advanced by distributing a theoretically very long required sequence onto shorter oligomer segments. Software is designed for encoding as well as decoding of messages, the latter being achieved via MS/MS analysis, which can be automated using the software. Ultimately, the system is demonstrated to work on the example of a QR code. In summary, this is a very nice piece of work, which should be published in Nature Communications after minor revisions as noted below.

- The authors indicate long term stability and zero energy storage of their data (i.e. molecules). Since no experimental proof is given in this direction, both claims should be removed.
- The term encoding and decoding are used inconsistently in the abstract and introduction, please correct.
- For reproducibility as well as for further work, the authors should give more details on the automated MS analysis. What is considered to be interpretable signal in MS/MS (signal to noise, error in m/z, baseline intensity, ...)?
- Kekulé did not discover benzene, he was the first to understand its structure and make the corresponding proposal for a structure.
- In the first encoding and decoding example, "human readable" oligomers are designed by using one acrylate (i.e. one defined side group of the oligomers) per to be written letter (and the words, one per oligomer, are coded by numbers to re-establish a sentence) – the authors should discuss the limitations of this system [i.e. an alphabet is encoded, but not all letters (i.e. all characters of the usually used ASCII code) can be written].
- In the example above (human readable), but even more in the QR code encoding/decoding, the authors should make some comment on the physical appearance of the oligomers. Are they analysed as a mixture, or, presumably, as individual oligomers as obtained after the synthesis? If the latter is the case, the claimed high data density is unfortunately not reached (yet), but the system has only the potential to reach it (i.e. the 71 oligomers have to be stored in individual sample vials, have to be analysed 1 by 1, ...). This should be discussed and mentioned. Most importantly, an outlook on a practical system should be given.
- At least all novel acrylates (or even better all synthesized acrylates) require HRMS to complete the characterization (SI).
- It would be very beneficial to provide a purity estimation of the prepared oligomers – the authors have, in a best practice manner, included an intensive set of characterization data, including chromatographic data, which should allow for such an estimation (maybe as addition to Supplementary Table 1).
- Some of the rather bold claims made in the conclusion should maybe be a bit more reflected.

Reviewer #2 (Remarks to the Author):

This article by Du Prez and coworkers describe the use of sequence-defined oligo(amide urethanes) as information-storage macromolecules. These polymers were prepared by solid-phase synthesis using a library of 20 building-blocks. This alphabet allows encoding of text as well as QR codes in libraries of sequence-defined molecules. Two softwares Chemcoder and Chemreader were developed to encrypt and decode the polymers. This article is not the first one in the field. Among other articles, the synthesis and decoding of sequence-defined polymers has been already reported in Nature Communications by Lutz (Nat. Commun. 2015), Barner-Kowollik (Nat. Commun. 2016) and Meier (Nat. Commun. 2018) (all prior art is accurately cited in the

manuscript). Yet, this topic is underexplored and the present article brings interesting new features to the field; such as the use of an encryption software and base-20 coding. Thus, I believe that this text could be interesting for a broad audience of readers. Yet, in my opinion some revisions (listed below) are needed before the article can be published.

1- The use of the word "hard-disk" in the title of the manuscript is an oversell. The technology described in this paper has very little in common with a hard disk, as generally understood by the broad scientific community. Since this text is addressed to a multidisciplinary journal, such a title may confuse readers; in particular expert in informatics and data storage. The title shall be revised.

2- Lines 69-76, the relevance of alphabets containing several building-blocks for macromolecular storage has already been theoretically discussed by Lutz (*Macromolecules* 2015) and was recently demonstrated experimentally by Lutz (*Angewandte Chemie* 2018) and Meier (*European Polymer Journal* 2018). This prior art shall be cited in this section.

3- Read-out: it is not very clear why positive mode was chosen in this work for polymer sequencing. It has been shown that negative mode is also useful for clean oligourethanes sequencing (see Lutz/Charles, *International Journal of Mass Spectrometry*). The authors shall add some comments about how different MS techniques (i.e. ESI and MALDI) and different ionization modes could broaden their approach.

4- Lines 151-152: the sentence written in the text "To write or not to write oligos" is not the one that was encoded in Figure 3. It contains a single capital letter and lowercase letters, whereas only capital letters are used in Figure 3". If the current version of Chemcoder is restricted to capital letters, it shall be specified and in that case the sentence in lines 151-152 shall be written in capital letters.

5- Lines 175 and 238, the authors wrote that the encryption of a QR code is an "ambitious challenge". Using standard coding principles, it is not more challenging to encode an image, a QR code or a video than a text. In all cases, information is decomposed in a standard bit string. Inputting images (see Balasubramanian, *Angewandte Chemie* 2016) and films (see Church Science 2017) in macromolecules has already been demonstrated. In the present case, what was challenging is that 71 oligomers were used. In other words, the number of oligos was a challenge but not what was encoded in it (it could have been whatever). This shall be rephrased.

6- The use of molecular addresses to identify libraries components with ChemReader presents some analogies with the mass-tags recently developed by Lutz and Charles (*Nature Communications* 2017) for identifying bytes in single-chains. It shall maybe be briefly mentioned.

7- There is not much information in the main text about how the libraries are stored and handled in the solid-state. The use of well-plates for decoding is only mentioned in the SI. Some clarification is needed.

Reviewer #3 (Remarks to the Author):

The manuscript introduces a method of storing and reconstructing decoded information using amide-urethane containing oligomers. While the reviewer agrees that this work is original and the results are solid, the reviewer is not convinced that there are overwhelming advantages over the DNA-based system when synthetic oligomers are employed for the data storage. In particular, the authors stated that "DNA holds some serious practical issues in terms of long-term stability and synthetic scalability". However, the authors failed to demonstrate how these issues can be eliminated in sequence-defined synthetic polymers (not just for oligomers). Therefore, the

reviewer suggests that the manuscript should not be published until the authors address this concern.

First of all, we would like to thank the reviewers for their valuable suggestions and detailed remarks, as a result of which we performed a number of changes to the main article and supporting info to meet the requirements of the reviewers. All implemented modifications have been indicated by yellow highlighting.

In what follows, we provide a point-by-point response to all comments made by the three reviewers.

Reviewer: 1

1) The authors indicate long term stability and zero energy storage of their data (i.e. molecules). Since no experimental proof is given in this direction, both claims should be removed.

We acknowledge that no direct proof was given for these claims with regard to our specific sequences; it was indeed meant as a general statement toward the typical stability/storage of polymer chains. In the abstract, the sentence “*Here we show the potential use of multifunctional sequence-defined macromolecules as a zero-energy, long-term data storage medium.*” has thus been modified with absence of indications to the claims.

2) The term encoding and decoding are used inconsistently in the abstract and introduction, please correct.

We acknowledge that there have been some inconsistencies, which we corrected now in the revised version.

3) For reproducibility as well as for further work, the authors should give more details on the automated MS analysis. What is considered to be interpretable signal in MS/MS (signal to noise, error in m/z, baseline intensity, ...)?

We acknowledge that these details were missing and we added them to the supporting info. Signals were considered as interpretable if the error in m/z was not higher than 0.02 and the signal-to-noise ratio had to be higher than 5. In Chemreader, this error, as a parameter of the software, can be adjusted at will. We set the value to 0.02 for our measurements, because this is the typical error in m/z for the used MS equipment.

4) Kekulé did not discover benzene, he was the first to understand its structure and make the corresponding proposal for a structure.

This is indeed a fully justified remark and we have remediated accordingly. In the revised version, we indicated that Kekulé made a proposal for the structure of benzene during his time in Ghent.

5) In the first encoding and decoding example, “human readable” oligomers are designed by using one acrylate (i.e. one defined side group of the oligomers) per to be written letter (and the words, one per oligomer, are coded by numbers to re-establish a sentence) – the authors should discuss the limitations of this system [i.e. an alphabet is encoded, but not all letters (i.e. all characters of the usually used ACSII code) can be written].

We fully agree that this system has some limitations and is only useful as a first proof of concept. We have added a few lines to the text before figure 3 to highlight the limitations of this system: “While encoding a human-readable sentence in sequence-defined polymers provided a first proof-of-principle to demonstrate the power of the Chemreader algorithm, the applied encoding scheme is not scalable to larger text fragments due to variable-length position encoding and to larger alphabet sizes (e.g. ASCII or Unicode) as separate acrylates are needed for all characters in the alphabet.”

6) In the example above (human readable), but even more in the QR code encoding/decoding, the authors should make some comment on the physical appearance of the oligomers. Are they analysed as a mixture, or, presumably, as individual oligomers as obtained after the synthesis? If the latter is the case, the claimed high data density is unfortunately not reached (yet), but the system has only the potential to reach it (i.e. the 71 oligomers have to be stored in individual sample vials, have to be analysed 1 by 1, ...). This should be discussed and mentioned. Most importantly, an outlook on a practical system should be given.

Indeed, for this research the oligomers have been analyzed one by one after the synthesis. We added a small text fragment before the conclusion to cope with this issue.

MALDI-TOF MS is capable to handle complex mixtures, as is widely demonstrated in proteomics where MALDI-TOF MS is used to characterize proteins by analysing multiple tryptic peptides consequently by MS/MS. This method is adopted in the 2D technique developed by the group of Jean-Francois Lutz (Angewandte 2016: Coding in 2D: Using Intentional Dispersity to Enhance the Information Capacity of Sequence-Coded Polymer Barcodes (<https://www.ncbi.nlm.nih.gov/pubmed/27484303>)). A complex sample is first analysed in the MS dimension. The different masses are subsequently isolated and analysed in a tandem MS. For this technique, the mass difference between different parent ions should be distinctive enough in order not to cause any problems during the isolation.

If the oligomer mixture becomes too complex, there are in fact two alternatives. The first is to start separating the oligomer mixture by Reversed phase HPLC and analyze the separated components by off-line (MALDI-TOF MS) or on-line (ESI-MS) mass spectrometry. The latter has the disadvantage that the spectral complexity is increased by the generation of multiple charged (fragments) ions. However, via deconvolution software these spectra could be transformed to single charged ion spectra that become amenable for our analysis software.

The alternative is the use of ion mobility mass spectrometry that allows for gas phase separation. However, this comes with a considerable additional instrument cost.

7) At least all novel acrylates (or even better all synthesized acrylates) require HRMS to complete the characterization (SI).

We acknowledge that these data were missing. Although HRMS measurements were performed for all acrylates, as requested, HRMS values could only be obtained for two of the five acrylates (2-ethoxyethyl and 2-acethoxyethyl). For the other three acrylates (propyl, heptyl and nonyl) we added LC chromatograms to compensate for the

absence of HRMS data (Supplementary Figures 4-6-8). The data are added in the SI to complete the characterization.

8) *It would be very beneficial to provide a purity estimation of the prepared oligomers – the authors have, in a best practice manner, included an intensive set of characterization data, including chromatographic data, which should allow for such an estimation (maybe as addition to Supplementary Table 1).*

We agree that this was missing in the supporting info. An extra column is added to Supplementary Table 1 in which estimations of the purities are given.

9) *Some of the rather bold claims made in the conclusion should maybe be a bit more reflected.*

While we strongly believe the original claims in the conclusion section, we adapted the conclusion section and added a section about the outlook/limitations:

“As the results obtained prove the possibilities for using these mono-disperse, multi-functional oligomers in the field of data storage, this study is another indication for the long-term potential that sequence-defined polymers hold to real-world applications and thus provides further validation for this rapidly developing branch of macromolecular chemistry. Undoubtedly, this will spark further research on the analysis and applicability of sequence-defined polymers worldwide. One of the main research challenges remains the further exploration of non-destructive techniques for the read-out of the sequence order in complex mixtures.”

Reviewer: 2

1) *The use of the word "hard-disk" in the title of the manuscript is an oversell. The technology described in this paper has very little in common with a hard disk, as generally understood by the broad scientific community. Since this text is addressed to a multidisciplinary journal, such a title may confuse readers; in particular expert in informatics and data storage. The title shall be revised.*

We acknowledge that the title could have been interpreted as the fact that we claimed all aspects of the construction of a ‘chemical hard disk’ in the manuscript. On the other hand, after consultation with the concerned IT-group, the term chemical hard disk is anyhow interpreted as a metaphor to attract a broader audience to the paper (as aimed for in Nature Communications) and is therefore not misleading.

We therefore changed the title into “Towards a chemical hard disk from multifunctional sequence-defined macromolecules”. The word ‘towards’ is then addressing the fact that we only describe certain aspects of what could be expected from a hard disk (data storage).

2) *Lines 69-76, the relevance of alphabets containing several building-blocks for macromolecular storage has already been theoretically discussed by Lutz (Macromolecules 2015) and was recently demonstrated experimentally by Lutz (Angewandte Chemie 2018) and Meier (European Polymer Journal 2018). This prior art shall be cited in this section.*

This is indeed a fully justified remark and we have remediated accordingly. In the revised version, we referred to the mentioned papers.

Lutz (Macromolecules 2015): reference 40

Lutz (Angewandte Chemie 2018): reference 31

Meier (European Polymer Journal 2018): reference 41

3) Read-out: it is not very clear why positive mode was chosen in this work for polymer sequencing. It has been shown that negative mode is also useful for clean oligourethanes sequencing (see Lutz/Charles, International Journal of Mass Spectrometry). The authors shall add some comments about how different MS techniques (i.e. ESI and MALDI) and different ionization modes could broaden their approach.

It is commonly known that MALDI-TOF MS signals (signal-to-noise ratio) are typically better in positive ion mode compared to negative ion mode. Since our preliminary data, obtained with positive ion mode were very promising, we did not investigate whether negative mode gives the same results. Although this mode is definitely an option, as proven by Lutz et al., we preferred positive mode because it works perfectly for both end groups (alcohol and acid). With positive mode we also observed that we could read from left to right and from right to left in the MS/MS spectra. This overlap from left to right and right to left helps with the analysis and read-out of the spectra. A benefit of the negative mode could be that only the fragments with an acid end group are visible in the MS/MS spectra, which would reduce the signals in the spectra and make them easier to analyse.

We choose for MALDI for two reasons.

- 1) To simplify the analysis of the mass spectra we choose MALDI over ESI, because with ESI multiple-charged fragments can occur which makes the spectrum more complex to analyse (we experienced this ourselves).
- 2) MALDI-MS/MS was permanently available to us. The ESI-MS/MS systems are extremely occupied for other applications (proteomics).

We are aware that ESI is easier to combine with separation techniques such as liquid chromatography, which could be useful in the future. Now, the oligomers are analyzed one by one on the MALDI-MS/MS, but if a combination between LC and ESI-MS/MS would be available, more complex samples, containing more oligomers, can be analyzed. The first separation would be done in the LC dimension and the analysis of the sequence order in the MS/MS dimension.

We added a small text fragment before Figure 1 to cope with this issue and added the mentioned reference: “Although both positive and negative ion mode proved to work in the past for a variety of sequence defined oligomers⁴⁸, only positive mode was used here because the signal-to-noise ratio of MALDI-TOF MS signals is typically better in this mode.”

4) Lines 151-152: the sentence written in the text “To write or not to write oligos” is not the one that was encoded in Figure 3. It contains a single capital letter and lowercase letters,

whereas only capital letters are used in Figure 3". If the current version of Chemcoder is restricted to capital letters, it shall be specified and in that case the sentence in lines 151-152 shall be written in capital letters.

We thank the reviewer a lot for this observation; we indeed forgot to take into account the difference between capital and normal letters. We have now put the sentence in capital letters, so that it corresponds to Figure 3 and will thus be clear to a broad audience.

*5) Lines 175 and 238, the authors wrote that the encryption of a QR code is an "ambitious challenge". Using standard coding principles, it is not more challenging to encode an image, a QR code or a video than a text. In all cases, information is decomposed in a standard bit string. Inputting images (see Balasubramanian, *Angewandte Chemie* 2016) and films (see Church Science 2017) in macromolecules has already been demonstrated. In the present case, what was challenging is that 71 oligomers were used. In other words, the number of oligos was a challenge but not what was encoded in it (it could have been whatever). This shall be rephrased.*

Indeed, we agree that a reformulation was necessary to clarify what the real challenge was for this part of the research. In both "Writing and reading of machine-readable data" and "Conclusion" sections, the text was therefore modified.

*6) The use of molecular addresses to identify libraries components with ChemReader presents some analogies with the mass-tags recently developed by Lutz and Charles (*Nature Communications* 2017) for identifying bytes in single-chains. It shall maybe be briefly mentioned.*

We agree that some analogies could be identified and we thus added the reference to the text.

7) There is not much information in the main text about how the libraries are stored and handled in the solid-state. The use of well-plates for decoding is only mentioned in the SI. Some clarification is needed.

We agree that we did not mention anything of the handling of the solid support in the main text, because we discussed this more in detail in a previous publication (<https://pubs.acs.org/doi/abs/10.1021/jacs.6b07120>). In the supporting info, starting at page S12, we discuss the handling of the solid support. The libraries, oligomers on the solid support, are stored in a fridge after synthesis. For the MS/MS-analysis, the oligomers are cleaved of the beads with a 1% TFA solution (page S12 bottom) and concentrated by evaporation and dissolved in THF for tandem MS analysis. We added a few lines to the supporting info to address these issues.

Reviewer: 3

The manuscript introduces a method of storing and reconstructing decoded information using amide-urethane containing oligomers. While the reviewer agrees that this work is original and the results are solid, the reviewer is not convinced that there are overwhelming advantages over the DNA-based system when synthetic oligomers are employed for the data

storage. In particular, the authors stated that "DNA holds some serious practical issues in terms of long-term stability and synthetic scalability". However, the authors failed to demonstrate how these issues can be eliminated in sequence-defined synthetic polymers (not just for oligomers). Therefore, the reviewer suggests that the manuscript should not be published until the authors address this concern.

In the upcoming section, we would like to clarify a couple of issues with regard to the general remark of the third reviewer.

The sequencing, i.e. the comprehensive deciphering from one chain end to the other, of the molecules of life, such as proteins and DNA, is by far the most significant biotechnological achievement of the 20th century. Consequently, current tools for sequence analysis of these natural macromolecules, supported by (i) elaborate standard operating protocols (SOPs) using commercial devices and (ii) well-documented analytical databases, readily allow for reading of the contained molecular data.

To date, molecular data management is attributed mostly to the ability of DNA as macromolecular carrier of information, able to archive, manage and encrypt data, which can be retrieved by established read-out tools. Moreover, the highest storage densities can be achieved, i.e. 10^6 times more information per mm^3 than in hard disk or flash memories. Despite the fact that the encoded information can be copied by DNA replication, water-soluble DNA holds some important disadvantages, which can clearly be overcome by implementation of man-made unimolecular sequences.

The major issue to emphasize in this context is that the structure of DNA is quite complex and the DNA 'alphabet' consists of the four-letter nucleotide code (A, C, G and T), which is very limited, compared to the vast diversity of (functionalized) repeating units in synthetic polymers. Moreover, synthetic carriers of digital information can have significantly simplified structures, regarding backbone and chirality. In fact, the simplest constructs potentially are atactic, sequence-controlled binary strings (0- and 1-bit) of information. Given the fact that no technology exists for limitless and flawless writing, resulting in very long (50 or more units) sequences, it should be stressed that storage capacity can be enhanced significantly by introducing tunable functional residues, both in identity and sequence order, resulting in more advanced information codes.

Other important issues, hampering the future large-scale use of DNA, are the limited amounts and stability of available DNA. The major argument concerning the limited amounts relates to the availability of biologically available phosphorus in nature (Childers et al. Bioscience 2011, 61, 117-124). Although recent research (<https://www.nature.com/articles/nmat4594#f1>) has indicated that a) DNA is more stable than flash memory, b) the amount of silicon will perhaps not be able to cope with the production of chips and c) DNA can be easily replicated, it should be emphasized that silicon in comparison to phosphorus is 300 times more available on earth and can be retrieved from more accessible minerals. Also, DNA replication is susceptible for errors.

While it is not feasible to produce DNA on a large enough scale, it has been proven recently (Mike Meier, KIT and Jeremiah Johnson, MIT) that simple sequence defined

oligomers can be made easily on multigram scale with cheap compounds, which shows the potential for further industrial upscaling. Besides hydrolysis, DNA is also sensitive to other degradation reactions such as deamination and dimerization. On the other hand, the backbone and side chains of a sequence defined oligomer and polymer can be chosen in such a way that it can cope with all these problems.

With regard to stability and cost of synthetic polymers, it can be noted that many types of polymers are known to remain quite stable over a very long period of time (multiple decennia to centuries) and their cost price is orders of magnitude lower than for DNA production, even for tailor-made structures such as a sequence-defined structures.

One last item is that protecting groups are needed during the synthesis of oligopeptides and oligonucleotides (as alternatives to DNA) while the thiolactone and other chemical platforms have proven that no protecting groups are needed for the synthesis of sequence defined structures.

We did not mention most of these issues in the text, because we assumed that many advantages and disadvantages of biological systems were known to the bigger audience. For sure, we are very well aware that still a lot of research needs to be done to outperform DNA, investigated for many decennia now, not only in terms of storage capacity but also with regard to the read-out tools.

REVIEWERS' COMMENTS:

Reviewer #1 (Remarks to the Author):

Du Prez et al. significantly improved the revised manuscript by carefully addressing all issues raised by the previous reviewers. In my point of view, the manuscript can now be published as is.

Reviewer #2 (Remarks to the Author):

The authors have significantly revised their manuscript and have taken into account all the comments of the three reviewers. The manuscript was improved and I am sure that it will be of great interest for the readers of Nature Communications.

However, I am still doubtful about the title of the article (first remark of my previous evaluation). Although I fully understand the arguments of the authors about choosing a catchy title, I think that the words "hard disk" shall be removed.

If reviewer 3 was fully right to mention that synthetic polymers are far to outperform DNA, I think that synthetic polymers are even more far to outperform current hard disk technologies. In fact, the performances of a hard disk are much more demanding than simply storing data. There are implications in terms of random access and processing speed that polymers will maybe never overcome. In this context, the results presented by the authors are not "towards" but "very far from" a chemical hard disk.

I recommend to use a more factual title like "Information-storage in multifunctional sequence-defined macromolecules" or something approaching. Such a title reflects better what the authors have really achieved in this work.

I think that the present work is very nice and original piece of chemistry and, in my opinion, a fancy title is not needed to convince the readers about its relevance. The data speak by themselves.

Reviewer #3 (Remarks to the Author):

Although the authors well-addressed the comments in their response letter, there is no further revision/addition to the main context regarding the advantages of synthetic oligomer based data storage compared to DNA system. Nature communication is a prestigious journal for broad audience, but not for the polymer community only. The statement on DNA-based method seems very unfair and misleading, at least to the reviewer. The reviewer will let the editor make the final judgement at this stage.

First of all, we would like to thank the reviewers again for their valuable suggestions and detailed remarks, as a result of which we performed a number of changes to the main article and supporting info to meet the requirements of the reviewers.

In what follows, we provide a point-by-point response to all comments made by the editor and the three reviewers.

REVIEWERS' COMMENTS:

Reviewer #1:

Du Prez et al. significantly improved the revised manuscript by carefully addressing all issues raised by the previous reviewers. In my point of view, the manuscript can now be published as is.

We thank the reviewer for the in-depth contributions and questions to improve the quality of the manuscript and for the confidence to publish our article.

Reviewer #2 (Remarks to the Author):

The authors have significantly revised their manuscript and have taken into account all the comments of the three reviewers. The manuscript was improved and I am sure that it will be of great interest for the readers of Nature Communications.

However, I am still doubtful about the title of the article (first remark of my previous evaluation). Although I fully understand the arguments of the authors about choosing a catchy title, I think that the words "hard disk" shall be removed.

If reviewer 3 was fully right to mention that synthetic polymers are far to outperform DNA, I think that synthetic polymers are even more far to outperform current hard disk technologies. In fact, the performances of a hard disk are much more demanding than simply storing data. There are implications in terms of random access and processing speed that polymers will maybe never overcome. In this context, the results presented by the authors are not "towards" but "very far from" a chemical hard disk.

I recommend to use a more factual title like "Information-storage in multifunctional sequence-defined macromolecules" or something approaching. Such a title reflects better what the authors have really achieved in this work.

I think that the present work is very nice and original piece of chemistry and, in my opinion, a fancy title is not needed to convince the readers about its relevance. The data speak by themselves.

We thank the reviewer for the in-depth contributions and questions to improve the quality of the manuscript and for the confidence to publish our article.

The title has been changed to a more factual one: "Multifunctional sequence-defined macromolecules for chemical data storage".

Reviewer #3 (Remarks to the Author):

Although the authors well-addressed the comments in their response letter, there is no further revision/addition to the main context regarding the advantages of synthetic oligomer based data storage compared to DNA system. Nature communication is a prestigious journal for broad audience, but not for the polymer community only. The statement on DNA-based method seems very unfair and misleading, at least to the reviewer. The reviewer will let the editor make the final judgement at this stage.

We modified the text to compare in detail the advantages of synthetic oligomer based data storage to the DNA system.

“Although the encoded information can be copied by DNA replication, susceptible to errors, DNA holds serious practical issues related to long-term stability and synthetic scalability.¹² Indeed, DNA is sensitive to both hydrolysis and other degradation reactions, such as deamination and dimerization. These issues could be overcome with synthetic sequence-defined polymers if the backbone and side chains are chosen wisely. The structure of DNA is also quite complex, and the four-letter nucleotide code (A, C, G, T) that makes up its ‘alphabet’ is limited compared to the vast diversity of synthetic building blocks. Another important issue hampering the large-scale use of DNA is the limited availability, the latter mostly connected with the scarceness of biologically available phosphorus in nature.¹³ Although recent research indicates that DNA is more stable than flash memory and that the amount of silicon might not be able to cope with the production of chips, it should be emphasised that, compared to phosphorus, silicon is 300 times more easily available on earth and can be retrieved from more accessible minerals.³ While producing DNA on a large enough scale is not feasible, it has recently been proven that sequence-defined polymers can be made on multigram scale with cheap compounds, which shows the potential for further industrial upscaling.¹⁴⁻¹⁶ In addition, many types of polymers are known to remain fairly stable over a very long period of time (decennia to centuries), and their cost is significantly lower than that of DNA production, even for tailor-made structures.”